# *In vitro* and *in vivo* evaluation of *Ulva lactuca* for wound healing

**Chien-Hsing Wang[1,2], Zih-Ting Huang[3], Kuo-Feng Tai[ID][4]***

**1** Department of Surgery, Hualien Tzu Chi Hospital, Hualien, Taiwan, **2** School of Medicine, Tzu Chi University, Hualien, Taiwan, **3** Department of Nursing, Tzu Chi University, Hualien, Taiwan, **4** College of Nursing, Divisions of Basic Medicine, Tzu Chi University, Hualien, Taiwan

\* taikuofeng@gmail.com

**Data Availability Statement:** All relevant data are within the manuscript.

**Funding:** • Initials of the authors who received each award: Kuo-Feng Tai; Chien-Hsing Wang • Grant numbers awarded to each author: TCCT-1111A13 • The full name of each funder: Tzu Chi University •

## Abstract

*Ulva lactuca* (*U. lactuca*) is an important seaweed species. Some ingredients in this species are thought to accelerate wound healing. However, limited data on the use of seaweed for wound healing exists. This study examined whether ethanol or aqueous extracts of *U. lactuca* promote antioxidant and anti-inflammatory properties *in vitro* and wound healing *in vitro* and *in vivo*. Cell proliferation, antioxidation, and migration were observed in NIH3T3 cells treated with *U. lactuca* extract *in vitro*. Both *U. lactuca* extracts were examined for their ability to inhibit inflammatory cytokine synthesis in lipopolysaccharide (LPS)-stimulated RAW 264.7 cells. *In vivo* experiments involved four groups of albino mice (BALB/c; 10 mice per group). One 1.0 cm$^2$ wound was created via excision of full-thickness skin on the back of all mice. Group I mice were treated topically with the ethanol extract of *U. lactuca* (25 mg/mL) for 10 d. Group II mice were treated topically with an aqueous extract of *U. lactuca* (12.5 mg/mL) for 10 d. Group III mice received topical application of phosphate-buffered saline solution. Group IV mice wounds were maintained without treatment. Both extracts considerably increased fibroblast proliferation. The antioxidant activity of the *U. lactuca* extract was determined using a total antioxidant capacity assay. Both extracts inhibited the release of tumor necrosis factor-α (TNF-α) and interferon-γ (IFN-γ) from LPS-mediated inflammation in RAW 264.7 cells. These extracts also upregulated the expression of Th2 cytokines such as transforming growth factor beta 1 (TGF-β1) and interleukin 10 (IL-10) in RAW 264.7 cells under pro-inflammatory conditions. Both extracts enhanced the migratory ability of NIH3T3 cells. *U. lactuca* ethanol extract enhances wound healing properties *in vivo*. These results suggest that bioactive compounds derived from *U. lactuca* extract are beneficial for wound healing and anti-inflammatory therapies.

## Introduction

The skin is the largest organ of the body and covers its entire surface. When injured, it should repair quickly and perfectly, regardless of the size and severity of the injury, to prevent further harm [1]. Wound healing is divided into four stages: hemostasis, inflammation, proliferation, and maturation. After skin injury, blood extravasation causes platelet aggregation and

URL of each funder website: https://rnd.tcust.edu.tw/?Lang=zh-tw • Did the sponsors or funders play any role in the study design, data collection and analysis, decision to publish, or preparation of the manuscript? Data analysis.

coagulation, triggering an inflammatory response, which lays the foundation for repair [2, 3]. During the repair phase, epithelial cells, dermal fibroblasts, and vascular endothelial cells migrate, proliferate, and differentiate from adjacent uninjured tissues to the wound site, leading to the maturation phase [4].

Since ancient times, humans have empirically used many plant resources to treat wounds, including cuts, abrasions, and burns [5]. These medicinal plants are safe, reliable, clinically effective, and cheap [6], making them widely accepted by the public [6]. Seaweeds have been used in traditional medicine and the food industry for many centuries [7]. They grow widely along coasts and are readily available, prompting numerous studies to explore them as sources of novel therapeutic compounds [8]. Seaweeds contain bioactive compounds such as tannins, triterpenoids, and alkaloids, which can affect various stages of wound healing [9]. Seaweeds can prevent tissue damage and stimulate wound healing. Despite extensive research on medicinal plants for wound healing, limited data on marine resources exist, including seaweeds [8]. Developing convenient and low-cost wound dressings made from seaweed for wound healing is expected.

*Ulva lactuca* (*U. lactuca*), also known as green algae (sea lettuce), is a macroalga belonging to the phylum Chlorophyta. It contains secondary metabolites, such as saponins, triterpenoids, steroids, tannins, alkaloids, flavonoids, phenolic compounds [10]. Saponins and triterpenoids can accelerate wound healing and inhibit inflammatory reactions [11, 12]. Tannins and alkaloids can prevent bacterial growth [13]. Flavonoids and phenolic compounds possess antioxidant properties that reduce cell necrosis and tissue damage by repairing damaged blood vessels and decreasing lipid peroxidation [13, 14]. Premarathna et al. reported that the extract from *Sargassum ilicifolium* has stronger wound-healing properties than that from *U. lactuca* [8]. However, this finding is based on low-dose oral administration, and no evident therapeutic results have been demonstrated *in vivo*. *In vitro*, the aqueous extract of *U. lactuca* induced proliferation and migration of the mice fibroblasts compared to cells in the control group. Widyaningsih et al. reported that applying gels with 5% or 10% ethanol extract of *U. lactuca* topically has been shown to accelerate wound healing [15]. This is achieved by enhancing macrophage scores as well as increasing blood vessel density, epithelial thickness, and fibroblast counts. This study aimed to investigate whether ethanol or aqueous extracts of *U. lactuca* can promote antioxidant and anti-inflammatory properties and explore their wound healing via *in vitro* and *in vivo* studies. This study was conducted to explore the potential wound healing properties of the *U. lactuca* extracts. The *U. lactuca* extracts may present a potential therapeutic opportunity in wound healing.

## Materials and methods

### Preparation of *Ulva lactuca* ethanol extract

*U. lactuca* powder was provided by Kung-Long Ocean Biotechnology Co., Ltd. in eastern Hualien, Taiwan. Briefly, 100 g of dry *U. lactuca* powder was placed in 1000 mL 95% ethanol in a glass Erlenmeyer flask, stirred (300 rpm) at 60°C for 1 h, and mixed vigorously for 72 h at 25°C. The resulting mixture was filtered through Whatman No. 1 filter paper to obtain the ethanol extract. The supernatant was concentrated and dried using the CES-8080 freeze dryer (Panchum, Taiwan) (1,100 rpm, 37°C). The final concentrated *U. lactuca* extract was stored at -20°C until further use [16].

### Preparation of *Ulva lactuca* aqueous extract

*U. lactuca* was extracted using the hot water extraction method. Briefly, 100 g of dry *U. lactuca* powder was placed in 1000 mL sterile water in a glass Erlenmeyer flask, stirred (300 rpm) at

60°C for 1 h, and mixed vigorously for 72 h at room temperature. The mixture was filtered through Whatman No. 1 filter paper to obtain the aqueous extract. The extract was further concentrated and dried using the CES-8080 freeze dryer (Panchum) (1,100 rpm, 37°C). The final concentrated *U. lactuca* extract was stored at -20°C until further use [17].

## Cell culture

NIH3T3 and RAW 264.7 cells were purchased from the Taiwan Cell Line Bank (Bioresource Collection and Research Center, Food Industry Research). NIH3T3 is a mouse embryonic fibroblast cell line, and RAW 264.7 is a mouse macrophage cell line. Cells were grown in Dulbecco's modified Eagle's medium (DMEM; Seromed, Berlin, Germany) supplemented with 10% fetal bovine serum (Biological Industries, Israel), 2 mM L-glutamine, 100 U/mL penicillin, and 100 g/mL streptomycin in humidified air with 5% $CO_2$ at 37°C in an incubator (Panasonic Healthcare, Japan).

## MTT assay

Cell viability was determined using the 3-(4,5-Dimethylthiazol-2-yl)-2,5-diphenyltetrazolium bromide (MTT) assay, as previously described [18]. Briefly, cells were seeded at a density of $1 \times 10^3$ cells/well in a 96-well plate and cultured with DMEM for 16 h. Thereafter, cells were treated with serial concentrations of the ethanol (6.25, 12.5, 25 and 50 mg/mL) or aqueous extracts of *U. lactuca* (6.25, 12.5, 25 and 50 mg/mL) for 24 and 48 h. Treatments at each concentration were performed in triplicate. Thereafter, the medium was aspirated, and cells were washed with phosphate-buffered saline (PBS). Cells were subsequently incubated with MTT solution (5 mg/mL) for 4 h at 37°C. The supernatant was removed, and formazan was solubilized in isopropanol and measured spectrophotometrically at 570 nm (Hach DR900, USA). The mean absorbance of triplicate readings was calculated.

## Total antioxidant capacity test

The total antioxidant capacity of *U. lactuca* was detected using the CheKine™ Total Antioxidant Capacity (TAC) Assay Kit (KTB1500, Abbkine, CA, USA) [19]. Concentrations of 6.25, 12.5, and 25 mg/mL of the ethanol or aqueous extracts were evaluated for antioxidant activity. Briefly, 150 μL of substrate diluent and 15 μL of substrate were premixed. Thereafter, 15 μL of reaction buffer and 10 μL of samples were added. The mixture was subsequently incubated for 5 min at room temperature. Absorbance was measured at 593 nm using a microplate reader (Hach DR900, USA). All experiments were performed in triplicate (n = 3).

## RNA extraction and real-time reverse transcriptase-polymerase chain reaction

Real-time reverse transcriptase-polymerase chain reaction (Real-time PCR) was performed as previously described [20–22]. RAW 264.7 cells were incubated for 18 h with lipopolysaccharide (LPS) or vehicle in the presence or absence of *U. lactuca*. The total cellular RNA was isolated from cultured RAW 264.7 cells with the GENEzol™ reagent (Geneaid Biotech, New Taipei, Taiwan) and quantified using a spectrophotometer (Hach DR900, USA). Total RNA (5 μg) from each sample was reverse-transcribed (RT) using 5× RT buffer, 6 μL DTT, 2.5 μL dNTP (10 Mm), 1.5 μL oligo-(dT) primers, 1 μL RNase inhibitor (40 μ/μL), 0.75 μL and RTase (200 μ/μL), and 2 μL Moloney murine leukemia virus. The reaction was performed at 42°C for 1 h and 75°C for 10 min on the 2720 Thermal Cycler (Applied Biosystems, USA).

**Table 1. Primer sequences used in transcript level analysis for inflammation markers of RAW 264.7 cells.**

| Gene | Forward primer (5'–3') | Reverse primer (5'–3') |
|------|------------------------|------------------------|
| *GAPDH* | CAGCAACTCCCACTCTTCCAC | TGGTCCAGGGTTTCTTACTC |
| *IFN* | ATCTGGAGGAACTGGCAAAA | TTCAAGACTTCAAAGAGTCTGAGG |
| *TNF-α* | AGCCGATGGGTTGTACCT | TGAGTTGGTCCCCCTTCT |
| *TGF-β* | CTTCAGCTCCACAGAGAAGAACTGC | CACAATCATGTTGGACAACTGCTCC |
| *IL-10* | ACCAGCTGGACAACATACTGC | TCACTCTTCACCTGCTCCACT |

The reverse transcription products (5 μL) were mixed with Eco™ Real-Time PCR System (Illumina, San Diego, CA, USA) and primers, resulting in a final volume of 10 μL. Real-time PCR was performed using SimplyGreen qPCR Master Mix, Low Rox (GeneDireX, Inc., USA) under standard thermal cycling conditions: polymerase activation at 95˚C for 2 min, followed by 40 amplification cycles at 95˚C for 10 s and 60˚C for 30 s and a melt cycle at 95˚C for 15 s, 55˚C for 15 s, and 95˚C for 15 s. *GAPDH* was used as the reference housekeeping gene, and each primer master mix consisted of one forward and one reverse primer (10 μM each) and sterile water. The primer sequences are listed in Table 1.

### *In vitro* migration/wound healing assay

Cell migration was determined using the wound healing scratch assay [23]. NIH3T3 cells were inoculated into 6-well plates ($1 \times 10^6$ cells/well) and cultured for 24 h at 37˚C in 5% $CO_2$ to achieve 80–90% cell confluency. A scratch wound was created using a 200 μL pipette tip, and wound debris was washed away using PBS. Ethanol (25 mg/mL) or aqueous *U. lactuca* (12.5 mg/mL) extracts were added. The initial wounding and migration of cells in the scratched area after 12, 24, and 36 h were captured using a microscope (Nikon, Tokyo, Japan) and calculated using Image J software. Measurements were performed in triplicate and repeated independently three times. The scratch closure rate (%) was calculated as follows:

$$\text{Scratch closure rate (\%)} = (A_0 - A_t)/A_0 \times 100\%$$

where $A_0$ represents the scratch area at 0 h and $A_t$ represents the scratch area at the designated time point.

### Animals and mouse wound healing model

The excision model was established based on the ARRIVE (Animal Research: Reporting of In Vivo Experiments) guidelines (approval number 2022005, dated September 23, 2022) from the Institutional Animal Care and Use Committee of Tzu Chi University of Science and Technology. All animal care and experiments followed the Animal Care Guidelines of the Animal Center at Tzu Chi University. Syngeneic BALB/c female mice (6–8 weeks old) were housed under a 12-h dark/light cycle with free access to food and water. Forty syngeneic adult female mice were divided into four groups of ten mice each. Mice were anesthetized with 50 mg/kg sodium pentobarbital (Merck KGaA, Darmstadt, Germany) via intraperitoneal injection. A cutaneous excisional wound (1.0 cm$^2$) was created on their backs. Ethanol or aqueous *U. lactuca* extracts in 0.1 mL PBS were applied to the excisional lesions three times daily. Wound sizes were measured using calipers on days 3, 5, 9, and 12. The *in vivo* wound healing results were statistically analyzed using SPSS software. Mean expression was compared between groups using one-way analysis of variance (ANOVA, $p < 0.05$ was considered statistically significant). The *in vivo* experiments were conducted using four groups (10 in each) of albino mice (BALB/c; 10 mice per group). Mice in group I were topically treated with the ethanol extract of *U. lactuca* (25

mg/mL) for 10 d. Mice in group II were topically treated with the aqueous extract of *U. lactuca* (12.5 mg/mL) for 10 d. Group III mice received topical application of PBS solution, and group IV (control) wounds were untreated. Wound sizes were measured and calculated using Image J software. Changes in wound size were expressed as percentages of the original wound size [24]. Wound healing was calculated as follows: percentage of wound healing = (total wound area − present wound area)/(total wound area) × 100.

## Statistical analyses

Results were obtained from three independent experiments (n = 3), and data were expressed as means ± standard deviation. Student's *t*-test and one-way ANOVA, followed by the Bonferroni post hoc test, were used for comparisons between the two groups and multiple groups. Statistical significance was set at p <0.05.

## Results

### *Ulva lactuca* extract promotes cell proliferation

Cell viability was assessed using the MTT assay. As shown in Fig 1, the ethanol extract of *U. lactuca* promoted cell proliferation at concentrations in the range of 6.25–50 mg/mL, whereas the aqueous extract of *U. lactuca* promoted NIH3T3 cell proliferation at concentrations in the range of 6.25–12.5 mg/mL. However, the aqueous extract of *U. lactuca* inhibited cell proliferation at concentrations in the range of 25–50 mg/mL. Thus, the aqueous extract of *U. lactuca* was weakly cytotoxic to NIH3T3 at high doses. Consequently, in the following research, the highest concentrations used were 12.5 mg/mL for the aqueous extract of *U. lactuca* and 25 mg/mL for the ethanol extract *U. lactuca*.

### Antioxidant activity of *Ulva lactuca* extract

The total antioxidant capacity of *U. lactuca* was also examined, and the results indicated that the ethanol extract at 6.25 mg/mL to 25 mg/mL showed total antioxidant capacity in a dose-dependent manner. A similar trend was observed in the NIH3T3 cells treated with aqueous extract (Fig 2).

**Anti-inflammatory properties of *Ulva lactuca* extract.** To verify the anti-inflammatory activity of the *U. lactuca* extract, we explored whether the ethanol or aqueous extract protected

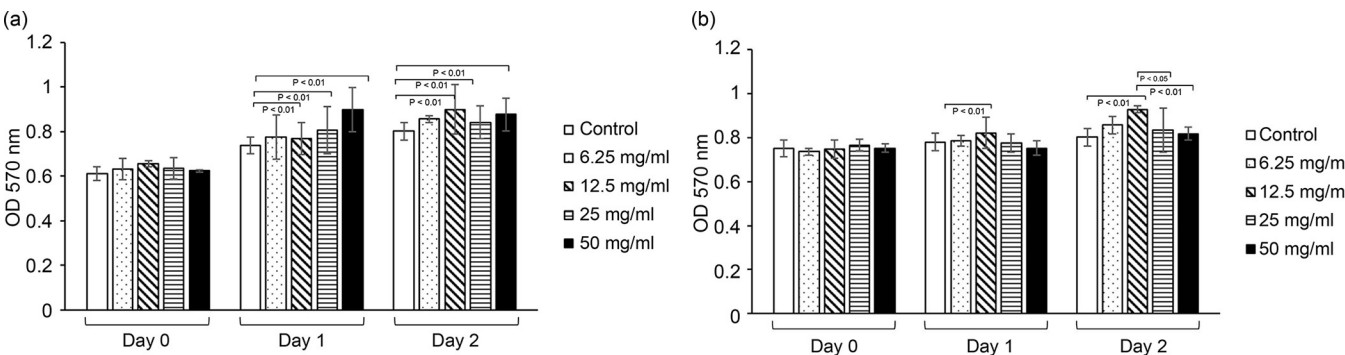

**Fig 1. *Ulva lactuca* increases the proliferation rate of fibroblasts.** One set of 96-well culture dishes was seeded with NIH3T3 cells. Cells were treated with serial concentrations of ethanol (A) or aqueous (B) *U. lactuca* extract for 24 h. At 24 h intervals, cell numbers were assayed using the MTT method. Cells were analyzed in triplicate for each sample. *p <0.01 vs. control. All experiments were performed in triplicates, and the results consistently showed the same trend. Data presented represent one of three independent experiments.

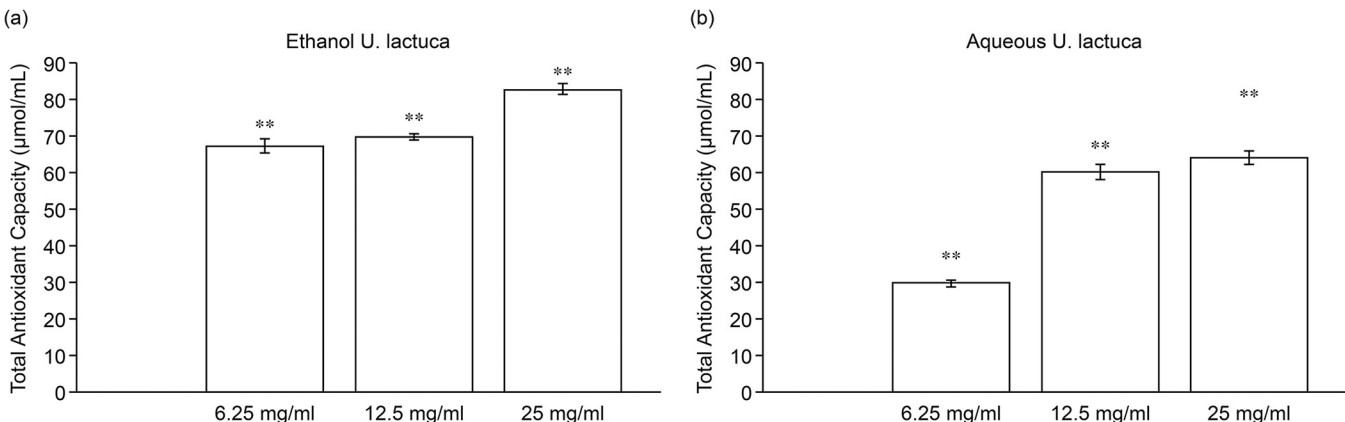

**Fig 2. *Ulva. lactuca* extract promotes antioxidant ability.** Total antioxidant capacity results for the *U. lactuca* extract using the CheKine™ Total Antioxidant Capacity (TAC) Assay Kit. Concentrations of 6.25, 12.5, and 25 mg/mL of the ethanol (A) or aqueous (B) extracts were evaluated for antioxidant activity. Absorbance was measured at 593 nm using a microplate reader. All experiments were performed in triplicates (n = 3). Values represent the mean ± S.D. Asterisks indicate statistical significance based on Student's t tests (**P <0.01).

against LPS-mediated inflammation in RAW 264.7 cells. Incubation of RAW 264.7 macrophages with LPS (1 μg/mL) significantly (p <0.01) increased Th1 cytokine (IFN-γ, TNF-α) overproduction and inhibited Th2 cytokine (IL-10, TGF-β) expression. However, the aqueous and ethanol extracts reduced Th1 cytokine expression and enhanced Th2 cytokine expression. Our results indicated that both aqueous and ethanol extracts of *U. lactuca* possess anti-inflammatory activities by suppressing the LPS-stimulated production of IFN-γ and TNF-α in RAW 264.7 cells treated with LPS (Fig 3).

### *Ulva lactuca* extract enhances cell migration

To assess the ability of *U. lactuca* extract to induce migration, a wound-healing scratch assay was performed *in vitro* using the 25 mg/mL ethanol extract or 12.5 mg/mL aqueous extract. Cell migration in NIH3T3 cells was photographed at 12, 24, and 36 h following extract treatment. As indicated in Fig 4A, cell migration on the slit of the confluent wells was photographed. An obvious enhancement of cell migration in NIH3T3 cells treated with aqueous or ethanol extracts was observed. A qualitative scratch assay was performed as indicated in Fig 4B. A significant enhanced effect (p <0.01) on cells treated with aqueous or ethanol extracts was observed (Fig 4B).

### Wound contraction effect in *Ulva lactuca*-treated mice

Quantitative measurements of wound size were routinely used to assess the initial wound size up to wound closure. Photographs and measurements of the wounds were taken on the day of wound creation (day 0) and on days 3, 5, 9, and 12 (Fig 5). The group treated with the ethanol extract of *U. lactuca* showed significantly reduced wound areas compared to the PBS treatment and control groups within three days (p <0.05). In the control group, the open wound area on the third day was 96.61 ± 8.21 mm$^2$. For the PBS-treated mice, it was 96.80 ± 9.32 mm$^2$, whereas for the mice treated with ethanol extract of *U. lactuca*, it was 77.63 ± 7.43 mm$^2$. The open wound area for mice treated with the aqueous extract of *U. lactuca* was 87.90 ± 8.62 mm$^2$. By the 5th day, wound areas were reduced in the groups treated with aqueous and ethanol extracts compared with those in the control group. The ethanol extract showed better

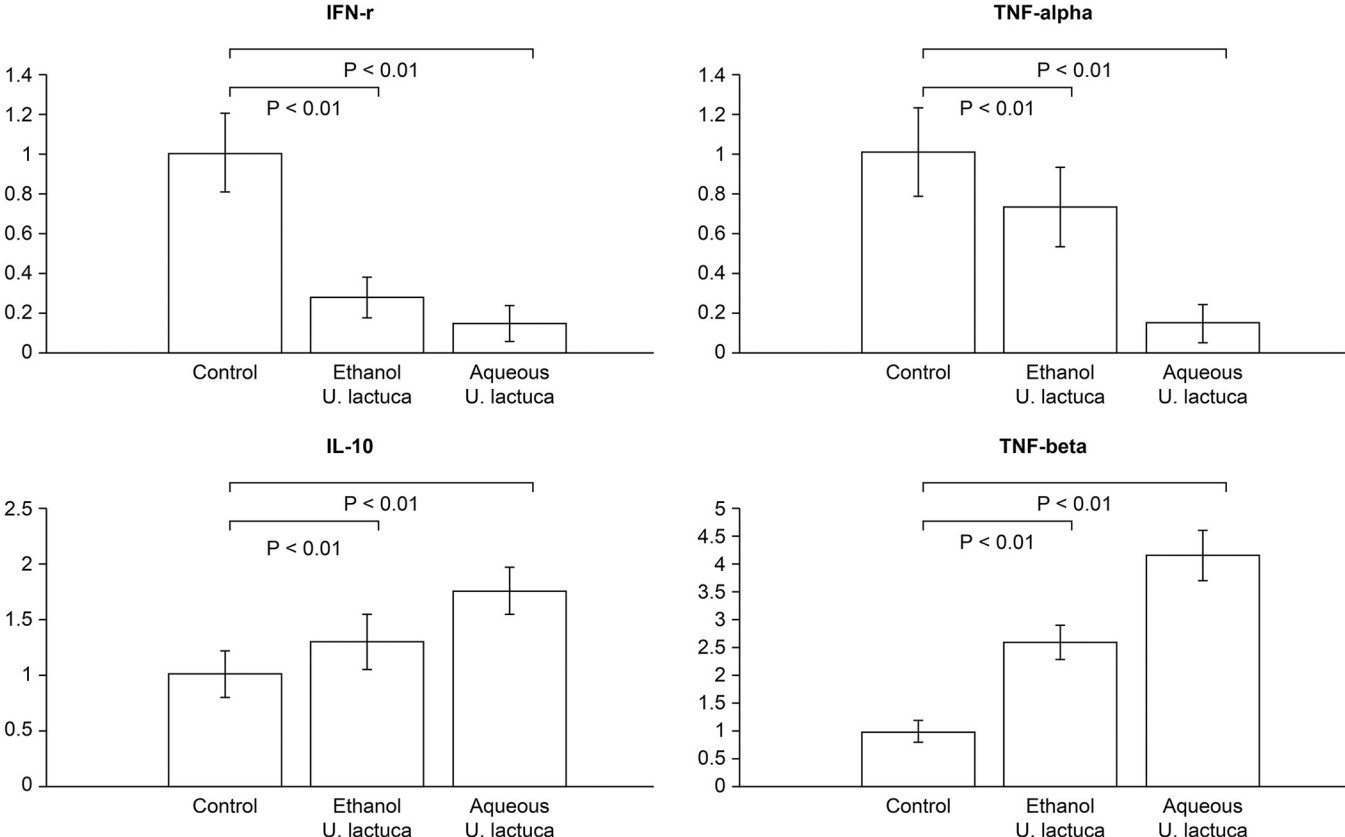

**Fig 3. *Ulva lactuca* inhibits mRNA expression of pro-inflammatory cytokines in RAW 264.7.** Quantitative analysis of mRNA levels of pro-inflammatory cytokines. The values from treated RAW 264.7 cells were normalized to match GAPDH measurements and subsequently expressed as a ratio to mRNA in the control cells (n = 4 per group). *p <0.01 vs. control.

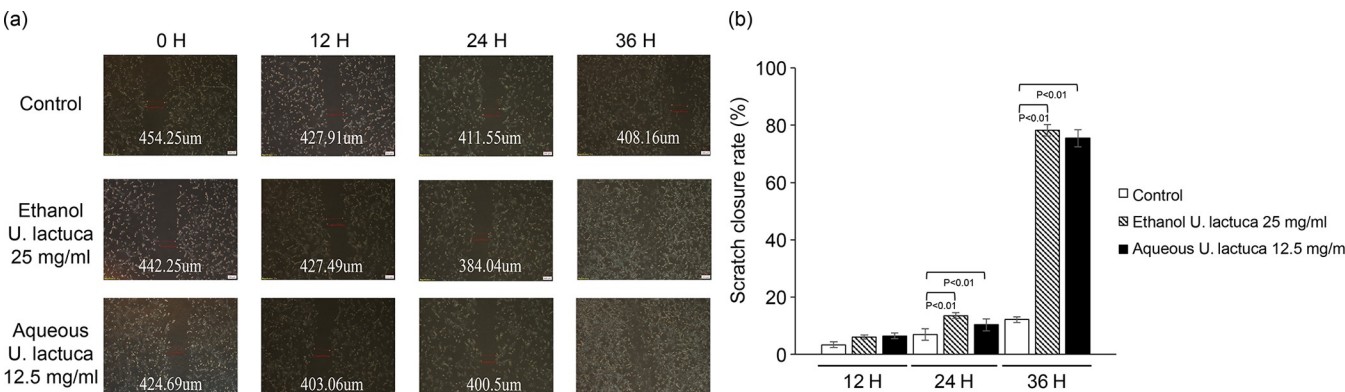

**Fig 4. *Ulva lactuca* increases NIH3T3 migration.** Representative image of NIH3T3 cell migration. *In vitro* wound healing assay. The scratched NIH3T3 cells seeded on 6-well plates were cultured in DMEM in the presence of 25 mg/mL *U. lactuca* ethanol extract or 12.5 mg/mL *U. lactuca* aqueous extract. The number of cells in the scratched areas increased differently among the four groups at 0, 12, 24, and 36 h. (A) The photographs are representative of three independent experiments. (B) A qualitative scratch assay was performed. The scratch closure rate (%) was obtained by averaging the results of three independent experiments. The bars represent the mean of three independent experiments. P <0.05 indicates that averaging differed significantly between two different groups (paired Student's t test). Scale bar represents 200 μm.

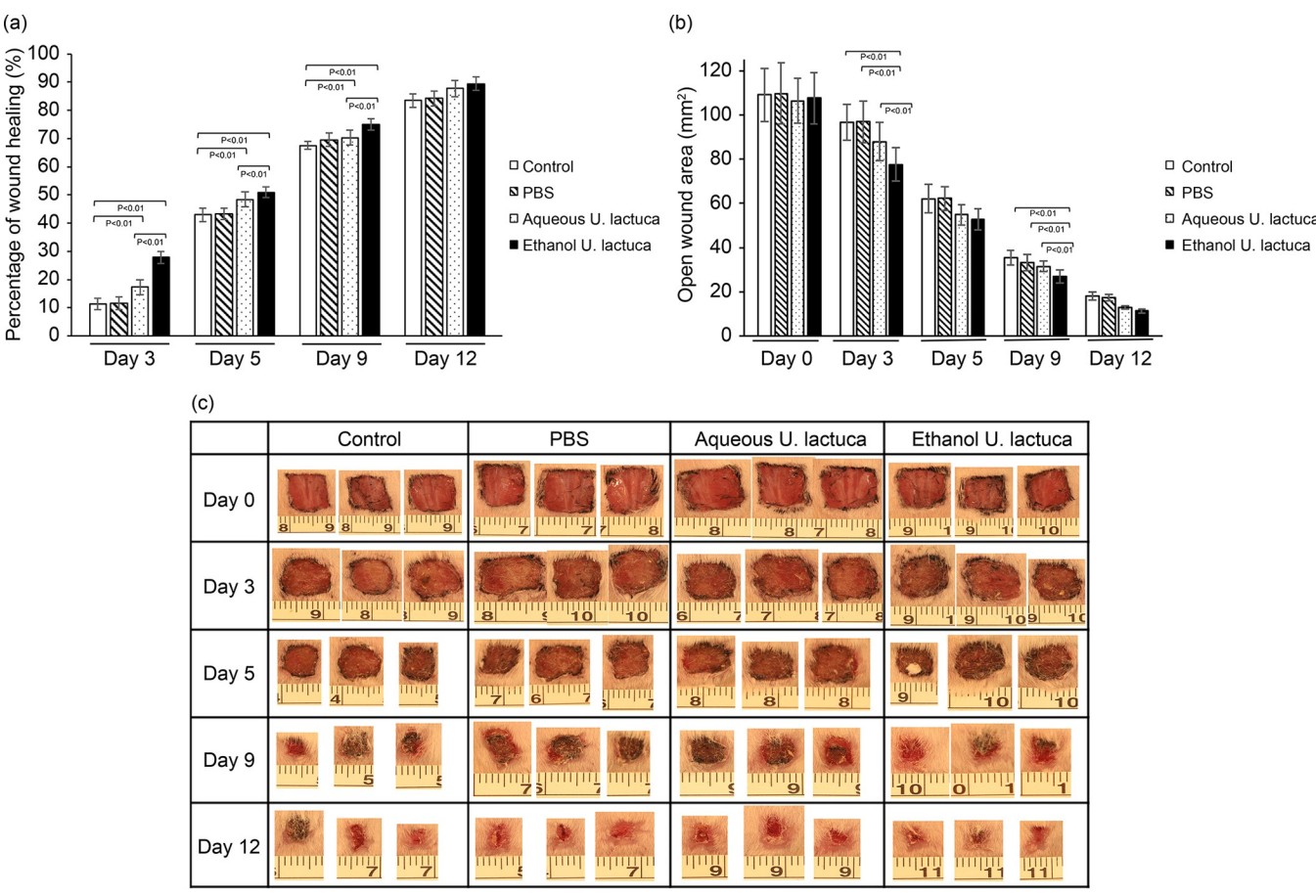

**Fig 5.** *Ulva lactuca* **extract treatment promotes wound contraction in BALB/c mice.** A 1.0 cm² cutaneous excisional lesion was created on the back skin of mice. Group I mice were treated with 25 mg/mL of *U. lactuca* ethanol extract three times daily. Excisional lesions of group II mice were treated with 12.5 mg/mL *U. lactuca* aqueous extract three times daily. Group III received PBS treatment, and group IV included control mice with untreated wounds. The size of the cutaneous lesions was measured using calipers on days 0, 3, 5, 9, and 12. Each group consisted of 10 mice. (A) Changes in the wound area at each time point relative to the original wound area of mice in each group over 12 d. The bars represent the percentage of wound healing in each group. p <0.05 indicates that the percentage of wound healing differed significantly between the two groups (paired Student's t-test). Percentage of wound healing = (total wound area − present wound area)/(total wound area) × 100. (B) Changes in wound area at each time point in each group over 12 d. (C) Digital photographs of mice at various stages of wound healing. The photographs show representative results of the wounds on days 0, 3, 5, 9, and 12 after wound creation. Photographs represent three out of ten mice.

wound healing activity than the aqueous extract on days 3 and 9. PBS treatment had no significant wound healing effect compared with the control group.

Ethanol extract therapy demonstrated better wound repair results than aqueous extract therapy. To confirm this, *in vivo* experiments were conducted using three groups of C57BL/6 mice (10 each). Mice in group I were treated topically with the ethanol extract of *U. lactuca* (25 mg/mL) for 9 d. Group II received PBS, and group III (control) received no treatment. The progressive healing changes in the wounds of the mice in each group are shown in Fig 6. Statistical analysis showed that by the end of the study period, treatment with the ethanol extract of *U. lactuca* caused a significant wound contraction compared to the PBS and control groups (p <0.01), indicating that the ethanol extract of *U. lactuca* possesses promising wound healing properties *in vivo*.

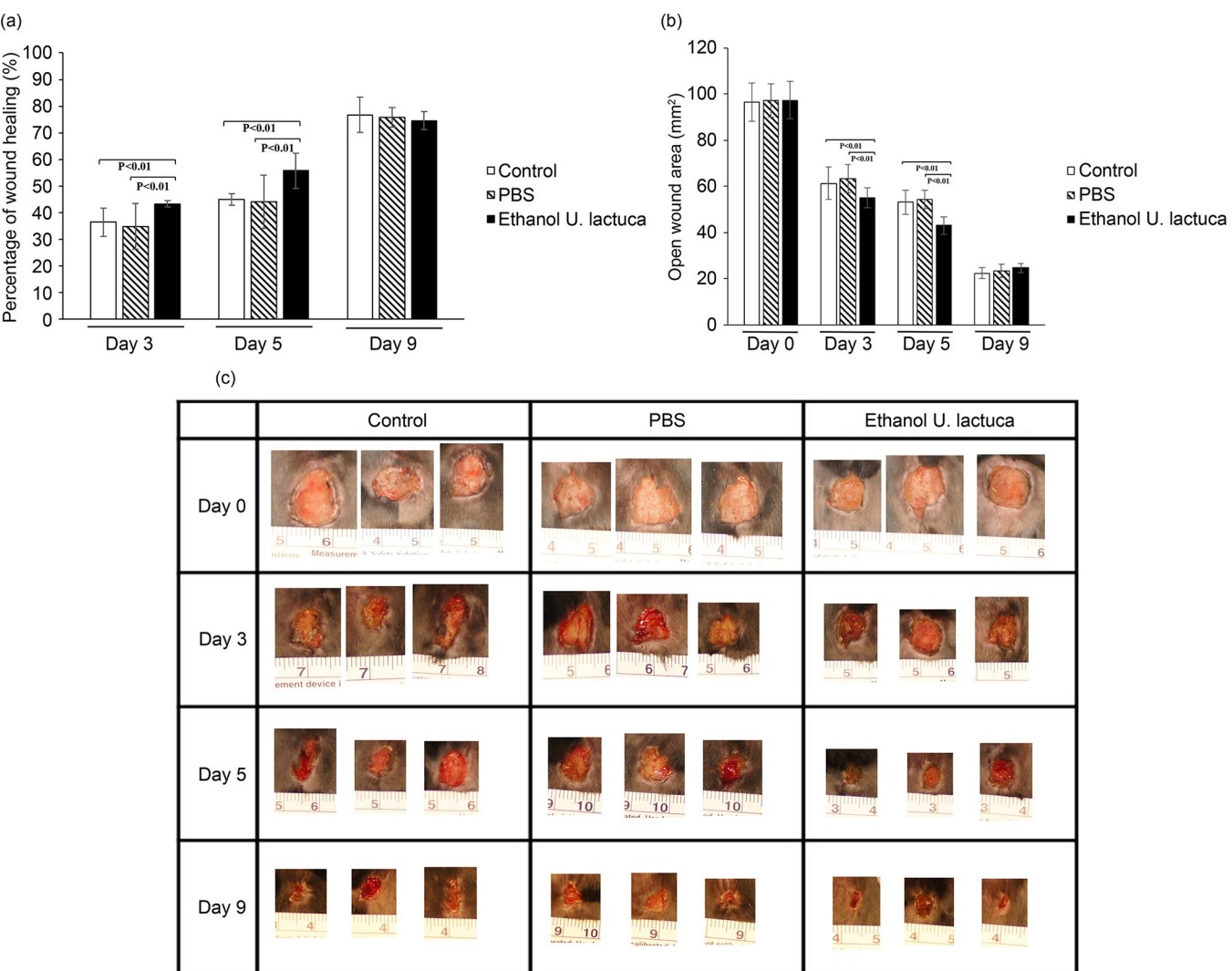

**Fig 6. The ethanol extract of *Ulva lactuca* promotes wound contraction of C57BL/6 mice.** A 1.0 cm$^2$ cutaneous excisional wound was created on the backs of mice. Group I mice were treated with 25 mg/mL of the *U. lactuca* ethanol extract three times daily. Group II mice were treated with PBS following excisional lesions. Group three comprised control mice with untreated wounds. Cutaneous wound size was measured using calipers on days 0, 3, 5, and 9. Each group consisted of 10 mice. (A) Changes in the wound area at each time point relative to the original wound area of mice in each group over 9 d. The bars represent the percentage of wound healing in each group. p < 0.05 indicates that the percentage of wound healing differed significantly between the two groups (paired Student's t-test). Percentage of wound healing = (Total wound area − Present wound area)/(Total wound area) × 100. (B) Changes in wound area at each time point of mice in each group over 9 d. (C) Digital photographs of mice at various stages of wound healing. The photographs show representative results of the wounds on days 0, 3, 5, and 9 after wound creation. Photographs represent three out of ten mice.

## Discussion

In this study, both ethanol and aqueous extracts of *U. lactuca* could reduce wound areas *in vivo*. Both extracts promoted antioxidation and anti-inflammation *in vitro*. The anti-inflammatory effects of *U. lactuca* demonstrated in this study are supported by studies showing that sulfated polysaccharides present in algae have anti-inflammatory properties [25]. A study by de Araújo et al. [26] showed that *U. lactuca* extract exerts anti-inflammatory effects by targeting bradykinin. According to Premarathna et al., the extract from *Sargassum ilicifolium* exhibits stronger wound-healing properties than that from *U. lactuca*. There are no evident therapeutic results of *U. lactuca in vivo*. However, Widyaningsih et al. found that topical

application of gels containing 5% or 10% ethanol extract of *U. lactuca* considerably accelerated wound healing [15]. In our study, wound areas were reduced in the group treated with aqueous and ethanol extracts compared to those of the normal wound control on the 5th day of treatment. The topical application of the ethanol extract of *U. lactuca* showed better wound healing activity than the aqueous extract on days 3 and 9.

The wound healing effects of *U. lactuca* are partly attributed to its rich polysaccharide content. A preliminary study indicated that the aqueous extract is richer in polysaccharides than the ethanol extract in *Ganoderma lucidum* extraction [27]. Hot water extraction is the most commonly used method for obtaining plant polysaccharides, being both easy to operate and suitable for industrial-scale applications. However, this method may result in polysaccharides containing soluble impurities. The addition of alcohol, acid, or alkali can enhance the purity of the extracted polysaccharides [28]. Ultrasonication, widely used for extracting Ulva polysaccharides, can further promote their dissolution [29]. In our experiments, the ethanol extract of *U. lactuca* demonstrated promising wound healing properties *in vivo*. A high concentration of the aqueous extract of *U. lactuca* (12.5–50 mg/mL) was marginally toxic to NIH3T3 cells. Therefore, we selected a low concentration (12.5 mg/mL) for animal experiments. Both the concentrated aqueous (12.5 mg/mL) and ethanol (25 mg/mL) extracts considerably increased fibroblast viability. However, we used a high concentration (25 mg/mL) of the ethanol extract in animal experiments. Although the ethanol extract showed better therapeutic effects, this may have been attributed to its higher concentration. Additional purification or removal of harmful substances from the aqueous extract of *U. lactuca* requires further investigation. Thus, the ethanol extract of *U. lactuca* may provide a potential therapeutic opportunity for wound healing. Investigating the combined use of alcohol and water extracts for wound healing might provide valuable insights into whether there are additive or synergistic therapeutic effects. This information would be valuable for optimizing wound treatment using *U. lactuca*.

## Conclusions

The present study revealed that both the aqueous extract and extracts of U. *lactuca* demonstrated better wound healing properties *in vivo* compared to the control group. Better therapeutic effects may be attributed to higher extraction doses or different extraction methods. Future studies should analyze the proportion of ingredients of the extracts from both alcohol and water extractions and remove impurities. Our study indicated that the ethanol extract of *U. lactuca* provides a new therapeutic approach for wound healing.

## Author Contributions

**Conceptualization:** Chien-Hsing Wang, Kuo-Feng Tai.

**Investigation:** Zih-Ting Huang.

**Methodology:** Zih-Ting Huang.

**Project administration:** Zih-Ting Huang.

**Writing – original draft:** Kuo-Feng Tai.

**Writing – review & editing:** Chien-Hsing Wang, Kuo-Feng Tai.

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
