## [Decision Letter · Decision Letter 0]

22 Oct 2024

PONE-D-24-40118In vitro and in vivo evaluation of Ulva lactuca for wound healingPLOS ONE

Dear Dr. Tai,

Thank you for submitting your manuscript to PLOS ONE. After careful consideration, we feel that it has merit but does not fully meet PLOS ONE’s publication criteria as it currently stands. Therefore, we invite you to submit a revised version of the manuscript that addresses the points raised during the review process.

We look forward to receiving your revised manuscript.

Kind regards,

Satheesh Sathianeson, Ph.D

Academic Editor

PLOS ONE

Journal Requirements:

2. To comply with PLOS ONE submissions requirements, please provide methods of sacrifice in the Methods section of your manuscript.

3. In your Methods section, please provide additional details regarding the plant material used in your study and ensure you have described the source. For more information regarding PLOS' policy on materials sharing and reporting, see https://journals.plos.org/plosone/s/materials-and-software-sharing#loc-sharing-materials.

For purposes of reporting, we request that you provide additional details as to the source of this material (please see http://journals.plos.org/plosone/s/criteria-for-publication#loc-3 for more information). Please provide the following details: If the plant material was purchased from a company or obtained from a third party, please provide data such as: name and location of the company/party supplying the product, a product description and name, a list of ingredients/compounds, product numbers, lot numbers, quality assessments, and chemical assessments.

“•          Initials of the authors who received each award：Kuo-Feng Tai; Chien-Hsing Wang

•            Grant numbers awarded to each author：TCCT-1111A13

•            The full name of each funder：Tzu Chi University

•            URL of each funder website：https://rnd.tcust.edu.tw/?Lang=zh-tw

•            Did the sponsors or funders play any role in the study design, data collection and analysis, decision to publish, or preparation of the manuscript? Data analysis”

“This study was supported by an Intramural Medical Research Grant from the Buddhist Tzu Chi College of Technology (TCCT-1111A13).”

5. We note that your Data Availability Statement is currently as follows: [All relevant data are within the manuscript and its Supporting Information files.]

6. PLOS requires an ORCID iD for the corresponding author in Editorial Manager on papers submitted after December 6th, 2016. Please ensure that you have an ORCID iD and that it is validated in Editorial Manager. To do this, go to ‘Update my Information’ (in the upper left-hand corner of the main menu), and click on the Fetch/Validate link next to the ORCID field. This will take you to the ORCID site and allow you to create a new iD or authenticate a pre-existing iD in Editorial Manager.

Reviewers' comments:

Reviewer's Responses to Questions

**Comments to the Author**

1. Is the manuscript technically sound, and do the data support the conclusions?

Reviewer #1: Partly

Reviewer #2: Yes

2. Has the statistical analysis been performed appropriately and rigorously? 

Reviewer #1: Yes

Reviewer #2: No

3. Have the authors made all data underlying the findings in their manuscript fully available?

Reviewer #1: Yes

Reviewer #2: Yes

4. Is the manuscript presented in an intelligible fashion and written in standard English?

Reviewer #1: Yes

Reviewer #2: Yes

5. Review Comments to the Author

Reviewer #1: The manuscript is about investigating wound healing, antioxidation, anti-inflation and cell profanation. There are a suite of tools that they used to reach their conclusion. Although the authors claimed that there is a lack in the information about the role of Ulva, they mentioned several similar studies in the Discussion. They should mention exactly in which ways their work is different

1- Line 34: should start with a capital letter

2- why did the authors used the phosphate-buffered saline solution instead of the ointments used normally for the injury to compare the effect with what is used normally?

3- it would be useful to have information about using the combination of the two extracts

4- several sentences in the introduction part need references to validate them, especially the first 3 sentences, and the line 70-73

5- In Material and Methods: Please mention the type of the flask the seaweed powder and the ethanol were put in, because at 60 ethanol will evaporate. Also mention the concentration of the ethanol used

6- line 167: remove the extra word “ethanol”

7- please justify the use of only female mice

8- Results: line 203: add “extract” after the aqueous

9- In the caption of figure 1, please add the number of replicates

10- Please add the scale for the bar on the images

11- Line 257-260: should be in the materials and the methods

12- It would be more informative if the photos were analyzed using image processing tool to validate the measurements

13- The Discussion section needs more information to elaborate further on the possible causes that enhanced the recovery. Which are the potential active compounds that could contribute to the healing?

14- The conclusion part needs to be reformulated and should be to the point giving the take-home message, and not repeating the results.

Reviewer #2: This study investigates the effects of the seaweed Ulva lactuca (U. lactuca) extracts on wound healing, encompassing both in vitro and in vivo experiments. The research topic is novel and holds practical value. The experimental design is mostly sound, and the data analysis methods are appropriate. However, several issues require further improvements.

1. The introduction lacks sufficient background information on U. lactuca. It is recommended to supplement it with more extensive literature reviews on its components and bioactivity to provide a stronger theoretical basis for the study.

2. In the results section, the statistical analysis for Figure 2 should include significance indicators in the figure legends. The results of the cell migration assay in Figure 4 should be quantitatively analyzed, providing data on migration rates.

3. It is suggested that the discussion section compares the findings of this study with other research on seaweed-promoted wound healing, highlighting similarities and differences.

Overall, while this study presents interesting findings on the wound healing properties of U. lactuca extracts, addressing the above points would significantly enhance the manuscript's quality and impact. The authors are encouraged to carefully consider these suggestions in their revision.

6. PLOS authors have the option to publish the peer review history of their article (what does this mean?). If published, this will include your full peer review and any attached files.

Reviewer #1: **Yes:**

Reviewer #2: **Yes: **

---

## [Author Response · Author response to Decision Letter 0]

13 Dec 2024

Response to Reviewers file uploaded in Attach files section.

---

## [Decision Letter · Decision Letter 1]

22 Dec 2024

In vitro and in vivo evaluation of Ulva lactuca for wound healing

PONE-D-24-40118R1

Dear Dr. Tai,

We’re pleased to inform you that your manuscript has been judged scientifically suitable for publication and will be formally accepted for publication once it meets all outstanding technical requirements.

Kind regards,

Satheesh Sathianeson, Ph.D

Academic Editor

PLOS ONE

Additional Editor Comments (optional):

Reviewers' comments:

Reviewer's Responses to Questions

**Comments to the Author**

1. If the authors have adequately addressed your comments raised in a previous round of review and you feel that this manuscript is now acceptable for publication, you may indicate that here to bypass the “Comments to the Author” section, enter your conflict of interest statement in the “Confidential to Editor” section, and submit your "Accept" recommendation.

Reviewer #1: All comments have been addressed

Reviewer #2: All comments have been addressed

2. Is the manuscript technically sound, and do the data support the conclusions?

Reviewer #1: Yes

Reviewer #2: Yes

3. Has the statistical analysis been performed appropriately and rigorously? 

Reviewer #1: Yes

Reviewer #2: Yes

4. Have the authors made all data underlying the findings in their manuscript fully available?

Reviewer #1: Yes

Reviewer #2: Yes

5. Is the manuscript presented in an intelligible fashion and written in standard English?

Reviewer #1: Yes

Reviewer #2: Yes

6. Review Comments to the Author

Reviewer #1: (No Response)

Reviewer #2: The authors have well addressed the reviewers' concerns. I have no other comments. The revised paper can be accepted.

7. PLOS authors have the option to publish the peer review history of their article (what does this mean?). If published, this will include your full peer review and any attached files.

Reviewer #1: **Yes:**

Reviewer #2: **Yes: **

---

## [Editor Report · Acceptance letter]

27 Dec 2024

PONE-D-24-40118R1 

PLOS ONE

Dear Dr. Tai, 

I'm pleased to inform you that your manuscript has been deemed suitable for publication in PLOS ONE. Congratulations! Your manuscript is now being handed over to our production team.

Kind regards, 

on behalf of

Dr. Satheesh Sathianeson 

Academic Editor

PLOS ONE